# Effects of Rinsing with Povidone-Iodine during Step II Periodontal Therapy: A Systematic Review and Meta-Analysis

**DOI:** 10.3390/jcm13072111

**Published:** 2024-04-04

**Authors:** Leonardo Svellenti, Jelena Karacic, Johannes Herzog, Moritz Tanner, Philipp Sahrmann

**Affiliations:** 1Department of Periodontology, Endodontology, and Cariology, University Center for Dental Medicine Basel UZB, University of Basel, 4058 Basel, Switzerland; leonardo.svellenti@unibas.ch (L.S.); jelena.karacic@unibas.ch (J.K.); 2Clinic of Conservative and Preventive Dentistry, Center of Dental Medicine, University of Zurich, 8032 Zurich, Switzerland

**Keywords:** inflammatory diseases, oral diseases, periodontitis, step II periodontal therapy

## Abstract

**Background and Objectives:** Antiseptics have been suggested to enhance the outcomes of non-surgical periodontal treatment (NSPT). Among these, povidone-iodine (PVP-iodine) was reported to significantly reduce periodontal pocket depths (PPDs). The aim of this study was to systematically re-assess the existing literature regarding the potential benefit of using PVP-iodine in step II periodontal therapy. **Material and Methods:** The scientific literature was systematically searched across electronic libraries for randomized controlled trials employing PVP-iodine during NSPT through to September 2023, using search items related to PVP-iodine and periodontitis. The selection process was performed by two independent reviewers. The primary outcomes were reductions in periodontal probing depth (PPD) and clinical attachment level (CAL). When reasonable, a meta-analysis of the included studies was performed. **Results:** Initially, 799 records were identified. After abstract and title screening and fulltext revision, 15 RCTs were included. The data from eight studies were merged in meta-analyses. The use of PVP-iodine had no significant effect on PPD reduction at 6 months (means [standard deviation]: −0.12 mm [−0.33; 0.09]) but it did at 12 months (−0.29 mm [−0.56; −0.02]). CAL was significantly better at 6 (−0.42 mm [−0.64; −0.20]) and 12 months (−0.39 mm [−0.66; −0.11]). **Conclusions:** PVP-iodine rinsing during NSPT may slightly improve patients’ PPD and CAL.

## 1. Introduction

Severe periodontitis is one major reason for tooth loss [1] and the sixth most common disease worldwide [2]. Health care costs caused by periodontitis are estimated at 90 billion euros based on data from the economically strongest countries of the European Union. Thus, periodontitis ranks third among all diseases from an economic point of view after diabetes mellitus and cardiovascular diseases [3]. A recent computing model, only for Switzerland, has shown that correct diagnosis and comprehensive professional periodontal treatment could save up to almost 7 billion Swiss francs and lead to significant benefits for patients [4].

The etiologic reason for periodontitis is biofilm on tooth surfaces, triggering the host’s immune response to release pro-inflammatory mediators. Consequently, collagen degradation results in the destruction of tooth-supporting tissues.

Non-surgical step II periodontal treatment [5] is a crucial step in the management of periodontitis [6,7,8]. Anatomical conditions in deep pockets and furcation areas, however, render a complete removal of biofilm difficult [9,10], thereby reducing the effectiveness of treatment [11] and rendering complete pocket elimination improbable [12].

Therefore, the topical use of antiseptics has been suggested to further reduce subgingival biofilm [13,14]. The available data from randomized controlled trials (RCTs) or comprehensive meta-analyses assessing a potential additional effect of antiseptics are limited. Even for chlorhexidine (CHX), often considered the gold standard of oral antiseptics, a review showed a remarkably inconsistent benefit on pocket depth reduction after daily use for 40–60 and 180 days, but not after 120 d, in cohorts with improper oral hygiene [15,16,17]. PVP-iodine has been shown to have potent antimicrobial properties that effectively control a wide range of periodontal pathogens [18,19,20]. When used as an adjunct, PVP-iodine can penetrate deep into the periodontal pocket and effectively reduce residual bacteria [11]. PVP-iodine was shown to have a small but significant effect on reducing pocket depths in 2010 [21].

The main advantage of PVP-iodine, in contrast to other antiseptics, is, therefore, attributed in the literature to its broad spectrum of activity in particular [18,20,22], while showing a low toxicity and few contraindications [23,24]. The small molecule iodine can easily penetrate microorganisms and lead to their cell death through the oxidation of various substances [25,26]. PVP-iodine showed a strong bacterial effect on clinical isolates even at a low concentration of 0.1% [22]. Its range of action covers Gram-positive and Gram-negative bacteria [20], a wide range of enveloped and nonenveloped viruses [27,28], spores [25] and fungal biofilms [29].

After a systematic review by our group more than 10 years ago had shown an additional benefit of the use of PVP-iodine [21], and taking into account the new randomized clinical trials published in the past few years, we felt that a careful update of the evidence for the use of PVP-iodine in step II periodontal therapy was necessary. Therefore, the aim of the present study was to re-evaluate the available literature and to assess the potential effect of the adjunctive use of PVP-iodine on the clinical outcomes of periodontal therapy.

## 2. Material and Methods

The study protocol was prepared in accordance with the PRISMA guidelines [30,31] (see Appendix A). The study was registered in the PROSPERO International Prospective Register of Systematic Reviews under ID number CRD42023393201.

### 2.1. Focused Question

The focused question of the present study was: Is there an effect of using PVP-iodine as a topical adjunct to step II non-surgical periodontal therapy in terms of periodontal probing depth reduction and improved clinical attachment level at 6 and 12 months, based on RCTs addressing the topic up to September 2023.

Studies were eligible for inclusion if they met the following PICOS criteria:Population: Adult patients with stage II–IV periodontitis or formerly classified as moderate–severe aggressive or chronic periodontitis.Intervention: Additional in-office use of PVP-iodine as an irrigation during non-surgical periodontal therapy/subgingival debridement.Comparison: Control group receiving water/saline irrigation or no irrigation during the respective treatment.Outcomes: Surrogate parameters of periodontal probing depth and clinical attachment level at six and twelve months.Study type: Randomized controlled clinical trials.

No distinctions were made regarding manual and ultrasonic instrumentation. Exclusion criteria were systemic diseases that could influence the prognosis of periodontitis or study protocols that did not include periodontal instrumentation or that included rinsing only after periodontal instrumentation had been employed. Cohorts with further adjunctive therapy like antibiotics or other antiseptic treatments were excluded from the review.

### 2.2. Study Outcome

Primary outcome was the assessment of periodontal probing depth and clinical attachment level at six and twelve months. Secondary outcomes were plaque index and bleeding-on-probing to provide a comprehensive assessment of periodontal health in response to these interventions.

### 2.3. Search Strategy

A literature search was conducted to retrieve relevant studies from the US National Library of Medicine (PubMed), the Excerpta Medical Database (Embase) and the Cochrane Central Library. The search included studies up to September 2023. Combinations of different keywords such as “PVP-iodine”, “iodine”, “PVP”, “povidone”, “periodontitis” and “periodontal” were used together with the corresponding MeSH terms. The search parameters used were identical for all three databases (for the precise search strategy see Appendix A).

Also, the reference lists of relevant articles were manually searched to identify additional RCTs.

### 2.4. Study Selection

In the first phase of this study, two independent reviewers (LS and JK) individually reviewed titles and abstracts retrieved by electronic search. All studies considered potentially eligible were then subjected to a full-text assessment. Any discrepancies regarding the inclusion of particular studies were resolved by discussion between the two reviewers and PS. Articles that were excluded after full-text screening are listed in Appendix A.

### 2.5. Data Extraction

Relevant data from the included studies, i.e., descriptions of the interventions in the different groups, size and further characteristics of the study cohorts, inclusion and exclusion criteria, study period and the reported surrogate parameters (see Table 1), were manually extracted by one author from the papers and were then documented in an Excel file.

### 2.6. Risk of Bias Analysis

In order to assess the study quality based on potential risks of bias of the included studies, the following parameters were analysed and rated, modifying the Grading of Recommendations assessment, Development and Evaluation (GRADE) [46]. For the strength of evidence, the studies were classified in four different degrees (high, medium, low and very low). Authoritative study characteristics for the strength of evidence are given in Table 2. Subcategories were rated by two independent reviewers with two, one or no point(s) for each study. Zero points were given if the parameter was not reported. One point was awarded for reporting without adequate explanation (i.e., randomization without description) and two points for exact description. The sum of these points categorized each individual paper as very low (<2), low (<6), medium or high quality (≥8 = high quality). Unequal ratings between the reviewers were resolved by discussion.

### 2.7. Statistical Analysis

Mean values and standard deviations for PPD and CAL reduction for different studies and assessment periods and the intergroup differences were calculated.

Meta-analyses of data from comparable studies were performed for the primary outcome variables if reported in at least two articles. Heterogeneity (I^2^) was used to express the percentage of total variation across the studies [47]. The GRADE guidelines were used to classify heterogeneity. Heterogeneity was rated as low for <40%, moderate for 30–60%, substantial for 50–90% and considerable for 75–100% [48].

Cohen’s d estimate of the average effect size [47] was performed. Forest plots were generated to illustrate effect sizes and funnel plots to detect publication bias. For the forest plots, fixed models were used. The *p*-value for statistical relevance was set at <0.05. Data analysis was performed using SPSS (IBM SPSS for Mac ver. 28.0, IBM Corp., Armonk, NY, USA) and the meta-analysis calculator Metamar version 3.5.1. [49].

## 3. Results

### 3.1. Literature Search and Screening

Initially, 799 records were identified from three different databases (see Figure 1). Duplicates were eliminated and titles and abstracts were screened with a reviewers’ accordance of a Cohen’s kappa of 88.7%. In a second step, the full texts of 24 studies were assessed for eligibility. The reasons for exclusion were no subgingival debridement (*n* = 1), only the existence of a conference paper (*n* = 4), missing control group (*n* = 1), no randomization (*n* = 2) and no irrigation during subgingival debridement (*n* = 1). During the full-text screening the reviewers’ accordance was 100% (24/24). Finally, 15 studies were included in the review, eight of which were included in meta-analyses. Five studies could not be included due to a very short a follow-up period (5 weeks to 3 months) [32,38,43,44,45], while in one study the mean values/standard deviations of the relevant parameters were not reported. Unfortunately, it was not possible to contact the authors to obtain this data [42]. In addition, one study could not be included in the meta-analysis because it was the only one that included patients with aggressive periodontitis [36].

### 3.2. Description of Included Studies

Table 1 depicts the main characteristics of the included RCTs. A total of 607 patients were included in all of the studies combined, with patient numbers varying from 12 [44] to 190 [13]. The oldest study was from 2001 [13], while the most recent was from 2021 [39]. Eight studies included patients with chronic periodontitis [33,34,35,40,41,43,44,45] and one included patients with aggressive periodontitis [36] according to the periodontal classification from 1999 [50], while one study included subjects with stage II/III and grade A/B periodontitis [39] according to the 2018 AAP/EFP classification of periodontal diseases [5], five did not provide any further information on the classification of periodontitis [13,32,37,38,42]. Seven of fifteen studies were performed in a split-mouth design [32,35,39,41,42,44], while the remaining eight studies had a parallel study design [13,33,34,36,37,40,43,45]. In terms of the average age of the subjects, most studies reported an average of between 40 and 55 years, while only the study assessing aggressive periodontitis reported an average age of less than 30 years [36]. Smokers were excluded from most studies. Only four studies did not report the smoking status of patients, while a single study included smokers [44]. For subgingival debridement, all studies used either an ultrasound device and/or hand instruments. The concentration of the PVP-iodine rinse varied from 0.1% to 10%, while in the greater part (9/15) of the studies the maximum concentration of 10% PVP was used. In all of the studies PVP-iodine was applied by the operator during subgingival debridement. In most cases application took place just once, but in two studies the irrigation with PVP was repeatedly applied [38,44]. In the control groups, NaCl was used in nine studies [32,35,36,37,39,41,42,43,45], tap water in five [13,33,34,40,44] and no irrigation was used in one study [38]. The investigation period varied from 5 weeks to 12 months, whereas all studies included in the meta-analysis had a follow-up period of at least 6 months.

### 3.3. Risk of Bias Analysis

In the studies reviewed, only three performed a sample size calculation to determine the minimal number of patients needed to address the hypothesis of the respective studies [36,39,43]. Concealment of the random allocation was described in nine out of fifteen studies. Randomization was further described in eleven studies, of which four used a table of randomization, three a computer-generated randomization, two a coin toss and two a sealed envelope. Examiner blinding was reported in five studies [36,37,40,43,44], while examiner calibration was performed in eleven studies. Thus, evaluation of these individual criteria resulted in four studies being high quality, two studies being medium quality, nine studies being low quality and no study was rated as having a very low quality (see Table 2).

In order to further visualize the studies identified with regard to their publication bias, a funnel plot was also created for the studies included in the meta-analysis (see Appendix A). This shows a symmetrical distribution, indicating a generally low publication bias.

### 3.4. Probing Depth

At baseline, the mean PPD values varied considerably from 2.77 mm to 7.11 (SD ± 1.45) mm in the studies included in the review with no significant inter-group. After 6 months, the range of the mean PPD values was 2.08 mm to 4.9 (SD ± 0.89) mm. The meta-analysis revealed no significant difference between the two groups after 6 months (mean: −0.12 mm [−0.33; 0.09]; *p* = 0.522, see Figure 2a), but after 12 months the use of PVP-iodine resulted in a significant additional effect in terms of PPD reduction (mean: −0.29 mm [−0.56; −0.02]; *p* = 0.037, see Figure 2b).

### 3.5. Clinical Attachment Level (CAL)

The average CAL for the studies included in the meta-analysis ranged from 3.1 to 9.5 (SD ± 2.26) mm and after 6 months from 1.8 to 8.23 (SD ± 2.03) mm, respectively. In both the test and control groups, the CAL values decreased after 6 and 12 months in all studies (see Figure 2c,d). A comparison between the two groups showed a significantly enhanced improvement by the additional use of PVP irrigation (*p* ≤ 0.001 after 6 months; *p* = 0.006 after 12 months). The mean inter-group difference at 6 months was 0.42 mm, with a range of 0.20 to 0.64 mm (95% CI), and after 12 months of 0.39 mm, ranging from 0.11 mm to 0.66 mm (95% CI), which were both in favour of the experimental group.

### 3.6. Effects on Secondary Outcome Parameters

The average plaque index at baseline varied between 4 and 99% (SD ± 24.8), and at the end of the study between 3 and 60% (SD ± 16.85), with only three of eleven studies having a plaque index greater than 20% in both groups at the end of the study [33,34,44]. No differences were observed between the two groups.

The average bleeding on probing at baseline varied between 28 and 89% (SD ± 18.9), and at the end of the study between 12 and 50% (SD ± 9.55). There were also no differences noted between the two groups.

## 4. Discussion

The aim of this study was to assess and update the existing body of evidence regarding a potential additional effect of povidone-iodine in reducing PPD and CAL in non-surgical periodontal therapy, addressing the lack of comprehensive meta-analyses of relevant studies over the past 15 years. PVP-iodine turned out to show no significant difference in probing depth after 6 months but did after 12 months, and CAL was in favour of an application of PVP-iodine after both 6 and 12 months.

According to the recent treatment guidelines of the European Federation of Periodontology [51], the use of adjunctive antiseptics, specifically CHX, may be considered. This recommendation primarily relies on a systematic review conducted by Da Costa et al. [15]. In this review, it was found that CHX demonstrated a small but significant improvement in PPD of 0.33 mm after 40 to 60 days and 0.24 mm after 180 days compared to the control group, but not at 120 days. Regarding CAL, no additional benefits were observed for CHX. Two important facts, however, should be considered regarding the included studies. First, CHX was used on a daily base for half a year and not only during subgingival debridement. Secondly, oral hygiene in both the test and the control groups was insufficient [52] at baseline and throughout the study period, with plaque index values ranging from 35 to 85%. In comparison, the greater part of the studies included in the present systematic review (7/10) reported sufficient oral hygiene with a plaque level below 25%.

Two studies included in the present review stood out as showing especially relevant effects of adjunctive PVP-iodine use: One study reported the combined use of PVP as an ointment and solution with the aim of maintaining a high concentration of SRP during the appointment. While ultrasonic scalers were used with povidone solution as a cooling liquid, povidone ointment, which in a clinical study showed a comparably high substantivity in the periodontal pocket, was re-applied into each pocket after the respective tooth had undergone subgingival instrumentation [53]. After 3 months, the reduction in PPD and recessions were found to be significantly higher as compared to the control group, which employed only water cooling during SRP. Thus, compared to baseline, the PPD values decreased by 3.1 ± 1.5 in the test group, whereas they decreased by only 2.3 ± 1.9 in the control group. The authors discussed that the reason for the pronounced benefit might be the maintenance of a high iodine concentration due to re-application and the ointment’s higher substantivity. Since the study was only followed-up for three months, this work did not find its way into the meta-analysis which assessed the effects after 6 and 12 months only.

In contrast to all of the other studies, one trial showed worse results when povidone-iodine was used with regard to PPD and CAL, even if the results failed to show significance [36]. In this study, full-mouth scaling was performed within a rather short time of 45 min. Furthermore, the study involved only patients (mean age: 28.56 ± 4.36 years) with a diagnosis of aggressive periodontitis. The authors mention the short rinsing and the fact that the rinse was not applied right up to pockets’ ground as possible reasons for the lack of an additional effect of PVP rinsing.

In addition to the risk-of-bias assessment, a funnel plot for the PPD values after 6 months was generated to visualize the potential risk of publication bias (see Appendix A), although the relevance is limited due to the small number of studies included. The funnel plot shows a generally symmetrical distribution, indicating a low publication bias with the exception of the study assessing patients with aggressive periodontitis [36].

A systematic review and meta-analysis from 2022 [54] which assessed the potential effect of PVP-iodine revealed an additional benefit of PVP-iodine in terms of PPD reduction only at 12 months after treatment, whereas there was no beneficial effect in PPD reduction after 6 months.

Regarding CAL, an improvement was only found at 6 months since there was only one trial assessing 12 months results. There are two important differences with the present study. Firstly, studies with a follow-up shorter than 6 months were included in the meta-analysis in that study, considering an interval which has been considered of low significance to the patient by recent EFP guidelines [51]. Secondly, several RCTs that met the published inclusion criteria were not included [37,38,40,42]. Thus, some published data contributing to the respective specific body of evidence have been lost from the analysis. While the reason why those studies were excluded in the full-text search or title/abstract screening is not clear, a possible reason might be the fact that the full texts of the missing studies were neither accessible for free nor available from the usual download option, thus being rather difficult to retrieve. Therefore, the authors consider that this systematic review can provide a new perspective on the effect of PVP-iodine in step-II periodontal therapy, as no known studies that met the inclusion criteria were excluded.

Some limitations of the present study could be the heterogeneity of some of the studies with regard to the PVP-iodine concentration as well as the varying differing PPD values. With regard to the PVP concentration, it must be mentioned that, with the exception of two studies [13,40], all of the studies that were included in the meta-analysis, used a PVP-iodine concentration of 7.5% or 10%. In contrast, the concentration used in the two studies mentioned above was 1% and 0.1%, i.e., almost one tenth and one hundredth of the concentrations used in the other studies. It therefore seems obvious that the effect of PVP-iodine in the respective studies is less powerful. Nevertheless, it must be mentioned, particularly with regard to clearance, that even high initial PVP-iodine concentrations are washed out and diluted after a short time (with saliva). It has been shown that the concentration of a 10% PVP-iodine rinse was reduced by around 50% after only 1 min and after 15 min only around 10% of the initial PVP-iodine concentration remained [55]. It can, therefore, be surmised that the concentration is certainly relevant; however, due to the rapid dilution of PVP-iodine [55], the far greater part depends on the type and above all the duration of application. In addition, it must be mentioned that a concentration of 0.1% PVP-iodine is already able to kill a large number of clinical isolates [22]. With regard to the reduction in PPD, a broad range was found between the different RCTs. These differences may be at least partially explained by the fact that different RCTs considered either full mouth PPD [43,45] or focused on individual teeth with higher initial values [34], thus showing a higher potential for reduction during healing [56]. A comparison of the significant PPD reduction of 0.12 mm after 6 months and 0.29 mm after 12 months shown in this study seems particularly relevant in relation to other adjuvant therapies. For example, a reduction of 0.485 mm was observed with the use of systemic antibiotics in long-term studies [57], while a reduction of 0.19 mm was observed with the use of locally delivered antibiotics for long-term studies [58]. The PPD reduction after 6 and 12 months in the present study are in the same range as those reported for locally delivered antibiotics [58]. This allows us to form the hypothesis that the effect of PVP-iodine could be equivalent or almost more effective than that found in the latter. Nevertheless, it must also be pointed out that another limitation of the study is certainly that only two studies were included after 12 months. Therefore, further studies investigating the long-term effects of PVP-iodine are certainly necessary.

It is important to place the small but significant differences observed resulting from additional use of PVP-iodine in non-surgical therapy in the overall context of periodontal therapy. Other factors, such as improving and maintaining efficient oral hygiene, realizing smoking cessation or the systemic use of antibiotics, have an either proven or supposedly greater additional effect. PVP-iodine, on the other hand, is characterized by the ease of its application, the low costs and its safeness in contrast to the immanent risks of bacterial resistance caused by antibiotics. In view of the above-mentioned limitations and with regard to the systematic review on CHX, PVP-iodine remains the only antiseptic that has a scientifically proven effect when used during subgingival debridement.

## 5. Conclusions

While the potential beneficial effect of PVP-iodine is not fully consistent over the different studies, there is–given the limitations of the systematic approach for the present review-a significantly enhanced CAL reduction when PVP-iodine is applied during SRP.

Thus, PVP-iodine can lead to improved clinical outcomes, suggesting its potential as a simple and cost-effective antiseptic in periodontal therapy.

## Figures and Tables

**Figure 1 jcm-13-02111-f001:**
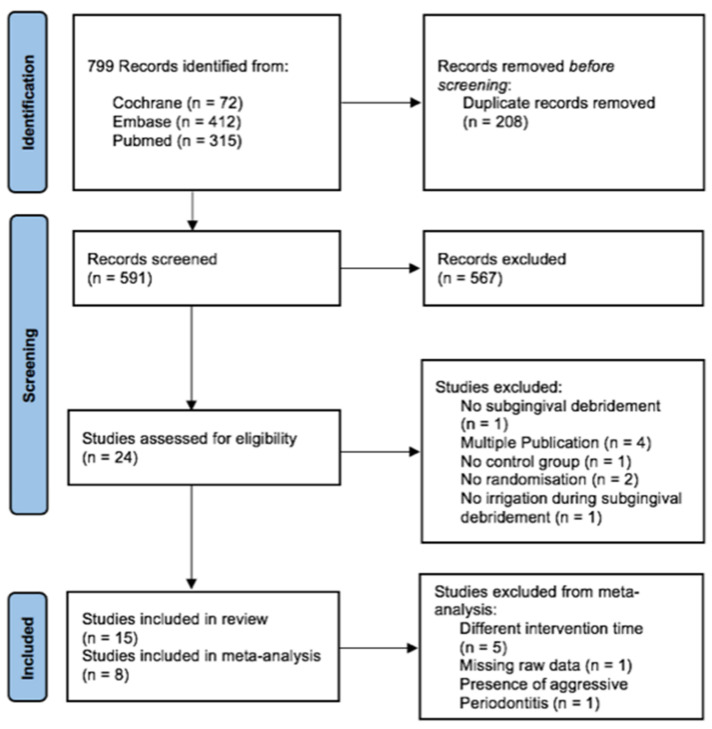
PRISMA flowchart of the database search and screening process.

**Figure 2 jcm-13-02111-f002:**
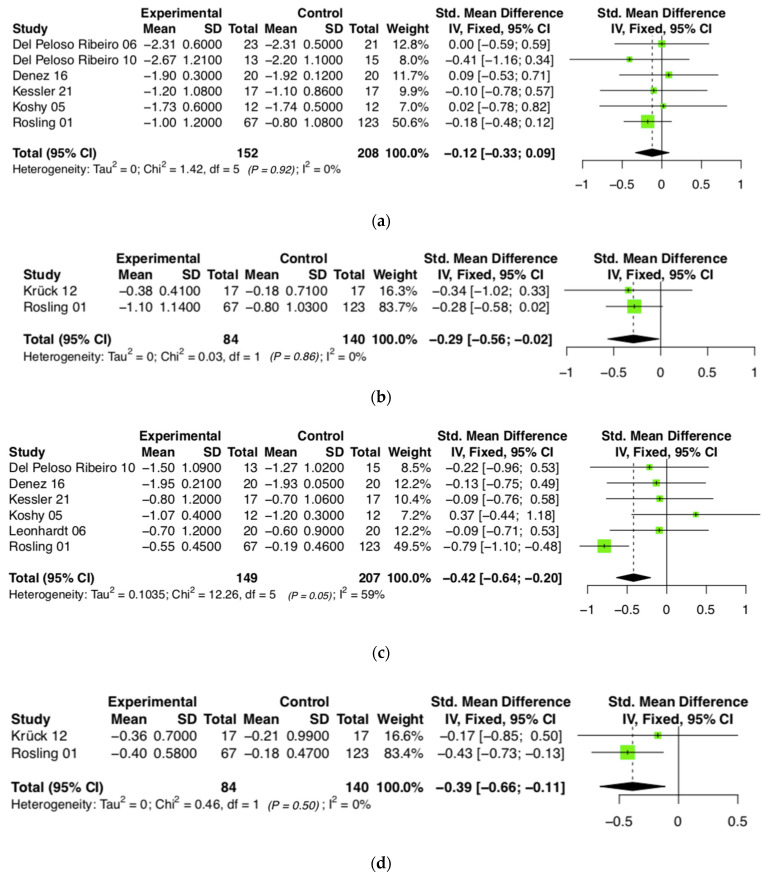
(**a**) Forest plot showing the comparison between the experimental group (irrigation with povidone-iodine) and the control group in terms of probing depth after 6 months. (**b**) Forest plot showing the comparison between the experimental group (irrigation with povidone-iodine) and the control group in terms of probing depth after 12 months. (**c**) Forest plot showing the comparison between the experimental group (irrigation with povidone-iodine) and the control group in terms of clinical attachment level after 6 months. (**d**) Forest plot showing the comparison between the experimental group (irrigation with povidone-iodine) and the control group in terms of clinical attachment level after 12 months [13,33,34,35,39,40,41,42].

**Table 1 jcm-13-02111-t001:** Description of included studies. BoP, Bleeding on probing; CAL, Clinical attachment level; FMBS, Full-mouth bleeding score; FMPS, Full-mouth plaque score; FMS, Full-mouth scaling; GI, Gingival index; NaCl, Sodium chloride; PAL, Probing attachment level; PPD, Probing depth; PGM, Position of gingival margin; PI, Plaque index; PVP-iodine, Povidone iodine; RAL, Relative attachment level; REC, Recession; RHAL, Relative horizontal attachment level; Rx bone loss, Radiological bone loss; US-Sc, Ultrasound scaler; VPI, Visual plaque index.

Author, Year of Publication	Population	Total Patients (after Drop Out)	Gender (% Female)	Mean Age (Years)	Intervention Test Group	Intervention Control Group	Clinical Parameters Assesed	Investigation Period (Months)	Follow-Up
Al-Saeed et al., 2009 [32]	Non-smokers, ≥ two sites per quadrant with ≥4 mm PPD and ≥2 mm CAL	16	38.00	42.90 (±7.55)	US-Sc + Irrigation with 1% PVP-iodine (Split-mouth design)	US-Sc + Irrigation with 0.9% NaCl (Split-mouth design)	BoP, CAL, PPD, PI	1.5, 3	Prophylaxis every 3 weeks
Del Peloso Ribeiro et al., 2006 [33]	Non-smokers, chronic periodontitis with Class II or III Furcation, BoP, ≥1 tooth with ≥5mm PPD	44	54.55	45.65	US-Sc + Irrigation with 10% PVP-iodine	US-Sc + Irrigation with distilled water	BoP, PPD, PGM, PI, RAL, RHAL	3, 6	Supragingival plaque control, oral hygiene instructions every 15 days, then 1×/month, periodontal instrumentation at 3-month reevaluation
Del Peloso Ribeiro et al., 2010 [34]	Non-smokers, chronic periodontitis, ≥1 molar with Class II Furcation, BoP, ≥5 mm PPD	28	53.25	43.35	US-Sc + 10% PVP-iodine as cooling liquid	US-Sc + water as cooling liquid	BoP, PPD, PGM, RAL, RHAL, VPI	1, 3, 6	Supragingival plaque control, oral hygiene instructions every 15 days, then 1×/month, periodontal instrumentation at 3-month reevaluation
Denez et al., 2016 [35]	Non-smokers, chronic periodontitis, ≥1 site per quadrant with ≥4 mm PPD, BoP, Rx bone loss	20	64.00	45.00	FMS (US-Sc and hand-instruments) + Irrigation with 10% PVP-iodine (Split-mouth design)	FMS (US-Sc and hand-instruments) + Irrigation with 0.9% NaCl (Split-mouth design)	CAL, GI, FMBS, FMPS, PPD, PI	1, 3, 6	Supragingival cleaning, oral hygiene instructions after 1, 3 and 6 months
Do Vale et al., 2016 [36]	Non-smokers, aggressive periodontitis, <35 years old, ≥8 teeth with ≥5mm PPD, BoP and ≥2 teeth with ≥7mm PPD	28	75.00	28.55	FMS (US-Sc) + 10% PVP as cooling liquid	FMS (US-Sc) + 0.9% NaCl as cooling liquid	FMBS, FMPS, PPD, PGM, RAL	1, 3, 6	1 Recall/ month, supragingival calculus removal and oral hygiene instructions when needed
Forabosco et al., 2006 [37]	Non-smokers, ≥7 teeth with ≥5 mm PPD, BoP	60	58.35	n/a	US-Sc + Irrigation with 10% PVP-iodine	US-Sc + Irrigation with NaCl	BoP, PAL, PPD	1, 3, 4	Periodontal instrumentation with groupwise irrigations every 30 days
Hoang et al., 2003 [38]	4 patients with Diabetes, 1 patient with psychiatric disease, 1 patient with down-syndrom, ≥1 site per quadrant with ≥6 mm PPD	16	43.75	n/a	Periodontal instrumentation (hand-instruments) of one quadrant + Irrigation with 10% PVP-iodine (Split-mouth design)	Periodontal instrumentation (hand-instruments) of one quadrant (Split-mouth design)	BoP, PPD, PI, REC	1.25	-
Kessler et al., 2021 [39]	Non-smokers, periodontitis stage II or III, grade A or B, ≥30 years old	17	47.05	51.80	FMS (US-Sc and hand-instruments) + Irrigation with 10% PVP-iodine (Split-mouth design)	FMS (US-Sc and hand-instruments) + Irrigation with 10% NaCl (Split-mouth design)	BoP, CAL, PPD, PI	1, 3, 6	Repeated periodontal instrumentation with groupwise irrigations at sites with ≥6 mm PPD after 3 and 6 months
Koshy et al., 2005 [40]	Non-smokers, chronic periodontitis, ≥2 sites per quadrant with ≥5 mm PPD, Rx bone loss	36	63.80	50.40	FMS (US-Sc) + Irrigation with PVP-iodine 1%	FMS (US-Sc) + Irrigation with distilled water	BoP, PAL, PPD, PI	1, 3, 6	Tooth cleaning, oral hygiene instructions every month
Krück et al., 2012 [41]	Chronic periodontitis, >30 years old	51	56.80	51.00	FMS (US-Sc and hand-instruments) + Irrigation with 7.5% PVP-iodine (Split-mouth design)	FMS (US-Sc and hand-instruments) + Irrigation with 0.9% NaCl (Split-mouth design)	BoP, CAL, PPD	3, 12	Supportive periodontal care by prophylaxis assistant every 3 months
Leonhardt et al., 2006 [42]	Non-smokers, ≥1 single-rooted tooth per quadrant with ≥6 mm PPD, BoP	20	60.00	54.00 (±7.55 years)	US-Sc + Irrigation with 0.5% PVP-iodine	US-Sc + Irrigation with sterile saline	BoP, PAL, PPD, PI	3, 6	-
Perrella et al., 2016 [43]	Non-smokers, chronic periodontitis, >30 years old, ≥1 site per quadrant with ≥5 mm PPD and ≥5 mm CAL	29	58.20	44.40	Periodontal instrumentation (hand-instruments) + Irrigation with 10% PVP-iodine	Periodontal instrumentation (hand-instruments) + Irrigation with 0.9% NaCl	BoP, CAL, GI, PPD, PI	3	Prophylaxis after 1 month, oral hygiene instructions, periodontal instrumentation after 3 months
Rosling et al., 2001 [13]	Non-smokers, ≥2 teeth per quadrant with ≥6 mm PPD	190	45.73	44.40	US-Sc + 0.1% PVP-iodine as cooling liquid	US-Sc + Tap water as cooling liquid	BoP, PAL, PPD, PI	3, 6, 12	Supportive periodontal therapy every 3–4 months, sites with ≥5 mm PPD and BoP received subgingival therapy with groupwise irrigations
Sahrmann et al., 2014 [44]	3/12 Smokers, chronic Periodontitis, ≥2 nonmolar teeth in the mandibula with ≥6 mm PPD	12	18.20	48.90	Periodontal instrumentation (US-Sc and hand-instruments) + Irrigation with 10% PVP-iodine solution and ointement (Split-mouth design)	Periodontal instrumentation (US-Sc and hand-instruments) + Irrigation with Tap water (Split-mouth design)	BoP, CAL, PPD, PI, REC	3	Periodontal instrumentation to teeth not included in the study were performed in the same session or after 4 to 8 days, supragingival debridement after 1 week, 1 month and 3 months
Zanatta et al., 2006 [45]	Non-smokers, chronic periodontitis, ≥8 teeth with ≥5 mm PPD, BoP	40	40.00	41.00	FMS (US-Sc) + Irrigation with 0.5% PVP-iodine	FMS (US-Sc) + Irrigation with 0.9% NaCl	BoP, CAL, PPD, PI REC	1, 3	Oral hygiene instructions, polishing every 2 weeks

**Table 2 jcm-13-02111-t002:** Risk of bias analysis. 0 = Not performed or no further information; 1 = Carried out with medium quality; 2 = Carried out with high quality. Overall score derived from total number: <2 = Very low quality; <6 = Low quality; ≥6 = Medium quality; ≥8 = High quality.

Author, Year of Publication	Sample Size Calculation	Concealment	Method of Randomisation	Blinding of Examiner	Examiner Calibration	Total Number	Overall Score
Al-Saeed 09 [32]	0	2	2	0	2	6	medium
Del Peloso Ribeiro 06 [33]	0	0	1	0	2	3	low
Del Peloso Ribeiro 10 [34]	0	0	1	0	2	3	low
Denez 16 [35]	0	0	2	0	2	4	low
Do Vale 16 [36]	2	2	1	2	2	9	high
Forabosco 06 [37]	0	0	0	2	0	2	low
Hoang 03 [38]	0	2	0	0	0	2	low
Kessler 21 [39]	2	0	2	0	2	6	medium
Koshy 05 [40]	0	2	2	2	2	8	high
Krück 12 [41]	0	2	2	0	0	4	low
Leonhardt 06 [42]	0	2	1	0	0	3	low
Perrella 16 [43]	2	2	2	2	2	10	high
Rosling 01 [13]	0	0	0	0	2	2	low
Sahrmann 14 [44]	0	2	2	2	2	8	high
Zanatta 06 [45]	0	0	0	0	2	2	low

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
