# Peer review of "Effects of Rinsing with Povidone-Iodine during Step II Periodontal Therapy: A Systematic Review and Meta-Analysis"

_jcm, 2024, doi:10.3390/jcm13072111_

Round 1

Reviewer 1 Report

Comments and Suggestions for Authors

Please check carefully that the guidelines for PRISMA 2020 were adhered to, even though the authors referenced it. For example:

1.       Regarding the search strategy – the full strategy should be detailed for Embase and Cochrane as well, not just PUBMED. This is in line with the PRISMA 2020 (see Item 7 of https://doi.org/10.1136/bmj.n160)

2.       Risk of bias anlaysis. How many reviewers assessed the study, multiple reviewers, process to resolve disagreement. (see Item 11)

3.       Please check the rest and comment here.

The included studies had a very wide range of PVP concentrations from 0.1 to 10%. This means that in some instances, patients were exposed to PVP 100 times weaker than 10%. This causes concern in how results were synthesised in the meta-analysis. Please kindly account for this.

Furthermore, the mean PPD (written as PD but should be termed as ‘PPD’ - for standardisation in the manuscript) varied from 2.77 to 7.11mm (Page 10 Line 88). Firstly, a mean PD of 7.11mm in patients is abnormally high. Please kindly confirm that this is correct. If it is, the range of 2.77 to 7.11mm is extremely diverse, and again raises concerns about the synthesised results in the meta-analysis.

On a related note, a mean PPD reduction of 0.29mm at 12 months is quite significant. For comparison, the Teughels review in the S3 guidelines PMID 31994207 reported a WMD of 0.485 when systemic antibiotics are used. When compared against locally delivered antibiotics, a WMD of 0.190 was found (Herrera review S3 guidelines). With the above mentioned issues, and the fact that only two studies were included for meta-analysis (12-month review), I wonder whether the 0.29mm is reliable.

Similarly, the authors mentioned in the discussion that there was a 2022 systematic review on the same topic. There were no included studies in this present review dated after 2022. Please kindly include a strong justification for this review, and discussion on any novel findings in this review.  

Other comments:

Abstract Line 14 – ‘in 2010’ – please remove. ‘PVP-iodine has been reported to significantly reduce PPD.’

In the introduction, I will like the authors to stronger justification the rationale for conducting a review on PVP-iodine. The authors mentioned that CHX is the gold standard for oral antiseptics, what does PVP-iodine bring that is unique and therefore could be used for clinical practice, instead of using CHX?

Line 88-89 – it may be more helpful to use PMPR or periodontal instrumentation, instead of scaling root planing, in line with S3 guidelines.

Regarding the search strategy – the full strategy should be detailed for Embase and Cochrane as well, not just PUBMED. This is in line with the PRISMA 2020 (see Item 7 of https://doi.org/10.1136/bmj.n160)

Line 111 – regarding excluded studies. I read through Supplemental Table II and noticed a lot of the studies were excluded because of ‘multiple publication of data from identical cohort’. What does this mean? If the authors are referencing another study that was included in the review, please state which studies were this in the supplement table in the ‘exclusion criteria’ column as well. Also, the full reference list of the excluded studies should be appended at the end of the document.

Line 115 – reference missing

Table 1. Gender percentage and mean age – should include SDs, if reported. Also, what strikes me is that some of the studies have disproportionate numbers in terms of gender. For example, Al-Saeed 09 has 38% female, and Sahrmann 14 has 18.2% females – why the disproportionate genders, especially if these were RCTs?

Page 6 Line 6 – reference missing

Page 6 Line 5 – for GRADE, there are four classes (high, moderate, low, very low). Why did the authors exclude ‘very low’ from the classification?

Table 2 – Concealement is a typographical error.

Regarding statistical analysis – some of the included studies were split-mouth studies – did the meta-analysis account for clustered data by comparing the variance within clusters (ie the same patient?)

Page 8 Line 18 – reference missing

Page 8 Line 29 – ‘one study lacked proper data reporting’. Were the authors of that study contacted for further information?

Page 9 Line 35 – reference missing

Page 9 Line 45 –the ‘classification of 2018’ is not very precise. It may be better to label it as the ‘2018 AAP/EFP classification of periodontal diseases’.

Page 10 Line 81, 92 – reference missing

I appreciate the funnel plot to assess publication bias, but with such a low number of studies included in this review, it is of limited use.

Author Response

The authors would like to thank the reviewer for the constructive criticism and the detailed and valuable feedback.

Comment 1:

Please check carefully that the guidelines for PRISMA 2020 were adhered to, even though the authors referenced it. For example:

  1. Regarding the search strategy – the full strategy should be detailed for Embase and Cochrane as well, not just PUBMED. This is in line with the PRISMA 2020 (see Item 7 of https://doi.org/10.1136/bmj.n160)
  2. Risk of bias analysis. How many reviewers assessed the study, multiple reviewers, process to resolve disagreement. (see Item 11)
  3. Please check the rest and comment here.

Response:

The authors thank the reviewer for the valuable comments. We have controlled the PRISMA guidelines again and implemented the amended items.

  1. As to the first comment, the full search strategy for Embase and Cochrane was added.
  2. The risk of bias analysis has been supplemented with the relevant information in order to describe it more precisely.
  3. Many thanks for the clarification of the PRISMA Guidelines and the attached link. We have gone through the document and have specified the “method to collect data” (as listed in item 9) and the number of reviewers involved under item 15 was added accordingly.

Revised text section:

  1. In chapter 2.3: “The search parameters used were identical for all three databases”
  2. In chapter 2.6: “by two independent reviewers” and “Discordant ratings between the reviewers were resolved by discussion”
  3. In chapter 2.5: “were manually extracted by one author from the papers and were then documented in an Excel file”

Comment 2: The included studies had a very wide range of PVP concentrations from 0.1 to 10%. This means that in some instances, patients were exposed to PVP 100 times weaker than 10%. This causes concern in how results were synthesised in the meta-analysis. Please kindly account for this.

Response:

Again, the authors thank the reviewer for the constructive criticism. We have addressed this limitation of the studies in the revised discussion and described this fact in more detail. The authors feel that that input helped a lot, especially with regard to the clearance of PVP-iodine.

Revised text section:

Page 14 Line 192: Limitations of the present study could be the heterogeneity of some studies with regard to the PVP-iodine concentration as well as the varying differing PPD values. With regard to the PVP concentration, it must be mentioned that with the exception of two studies (Koshy et al., 2005; Rosling, Hellström, Ramberg, Socransky, & Lindhe, 2001), all studies which were included in the meta-analysis, used a PVP-iodine concentration of 7.5% or 10%. In contrast, the concentration of the two studies mentioned above was 1% and 0.1%, i.e. almost one tenth and one hundredth of the concentrations used in the other studies. It therefore seems obvious that the effect of PVP-iodine in the respective studies is less powerful. Nevertheless, it must be mentioned, particularly with regard to clearance, that even high initial PVP-iodine concentrations, are washed out and diluted after a short time (with saliva). It has been shown that the concentration of a 10% PVP-iodine rinse was reduced by just around 50% after only 1 minute and after 15 minutes only around 10% of the initial PVP-iodine concentration remained (Sahrmann, Manz, Attin, Zbinden, & Schmidlin, 2015). It can therefore be surmised that the concentration is certainly relevant, however due to the rapid dilution time of PVP-Iodine (Sahrmann, Manz, Attin, Zbinden, & Schmidlin, 2015), the far greater part depends on the type and above all the duration of application. In addition, it must be mentioned that a concentration of 0.1% PVP-Iodine is already able to kill a large number of clinical isolates (Gocke et al., 1985).

Comment 3:

Furthermore, the mean PPD (written as PD but should be termed as ‘PPD’ - for standardisation in the manuscript) varied from 2.77 to 7.11mm (Page 10 Line 88). Firstly, a mean PD of 7.11mm in patients is abnormally high. Please kindly confirm that this is correct. If it is, the range of 2.77 to 7.11mm is extremely diverse, and again raises concerns about the synthesised results in the meta-analysis.

Response:

We would like to thank the reviewer for the detailed reading and the valuable comment. In the revised manuscript, we have adapted the abbreviation and used “PPD” throughout the text as suggested by the reviewer.

In fact, the range of the results from different RCTs is that large. In the revised discussion, we have now added this issue as a limitation.

Revised text section:

Page 14 Line 209: With regard to the reduction of PPD, a broad range was found between the different RCTs. These differences may be at least partially explained by the fact, that different RCTs considered either full mouth PPD (Perrella et al., 2021; Zanatta et al., 2006) or focused on individual teeth with higher initial values (Del Peloso Ribeiro et al., 2010), thus showing a higher potential for reduction during healing (Badersten et al., 1984). 

Comment 4: On a related note, a mean PPD reduction of 0.29mm at 12 months is quite significant. For comparison, the Teughels review in the S3 guidelines PMID 31994207 reported a WMD of 0.485 when systemic antibiotics are used. When compared against locally delivered antibiotics, a WMD of 0.190 was found (Herrera review S3 guidelines). With the above mentioned issues, and the fact that only two studies were included for meta-analysis (12-month review), I wonder whether the 0.29mm is reliable.

Response:

Many thanks again, especially for the reference mentioned. We have recontroled that value and found it confirmed.

The fact, that only two studies after 12 months could be analyzed certainly represents a limitation, which is addressed in the revised discussion section.

Revised text section:

Page 14 Line 214: A comparison of the significant PPD reduction of 0.12 mm after 6 months and 0.29 mm after 12 months shown in this study seems particularly relevant in relation to other adjuvant therapies. For example, a reduction of 0.485 mm was observed with the use of systemic antibiotics in long-term studies (Teughels et al., 2020), while a reduction of 0.19 mm was observed with the use of locally delivered antibiotics for long-term studies (Herrera et al., 2020). The PPD reduction after 6 and 12 months in the present study are in the same range as those reported for locally delivered antibiotics (Herrera et al., 2020). This could allow the hypothesis that the effect of PVP-Iodine could be equivalent or almost more effective than the latter. Nevertheless, it must also be pointed out that another limitation of the study is certainly that only 2 studies were included after 12 months. Therefore, further studies investigating the long-term effects of PVP-Iodine are certainly necessary.

Comment 5: Similarly, the authors mentioned in the discussion that there was a 2022 systematic review on the same topic. There were no included studies in this present review dated after 2022. Please kindly include a strong justification for this review, and discussion on any novel findings in this review.

Response:

Again, the authors thank for the constructive criticism and see the reviewer’s point clearly: Another systematic review on the same topic only few years before seems to render a new respective study questionable. We noticed, however, considerable flaws regarding the previous review which we were aware of when we started working on the present manuscript:

  1. The mentioned review did not consider studies that met the inclusion criteria. Since they listed the studies in the list of excluded studies, but no reason for exclusion is stated, we believe that they had difficulties finding the corresponding full texts. We have mentioned this in the discussion section of our manuscript.
  2. The focus of the present study is specifically on the effect of PVP-iodine. After more than 10y after publication of the previous review by our group we felt that a careful update of the evidence of using PVP-iodine in step 2 therapy is needed.

So, we have addressed these differences to the review from 2022 in the revised discussion and highlighting the justification for our review.

Revised text sections:

Page 13 Line 189: “Therefore, the authors consider that this systematic review can provide a new perspective on the effect of PVP-iodine in step-II periodontal therapy, as no known studies that met the inclusion criteria were excluded”

Page 2 Line 73: “After a systematic review by our group more than 10 years ago had shown an additional benefit of the use of PVP iodine (Sahrmann et al., 2010), and taking into account the new randomized clinical trials published in the past few years, we felt that a careful update of the evidence for the use of PVP iodine in step II periodontal therapy was necessary. Therefore, the aim of the present study was to re-evaluate the available literature and to assess the potential effect of the adjunctive use of PVP iodine on the clinical outcomes of periodontal therapy.”

Comment 6: Abstract Line 14 – ‘in 2010’ – please remove. ‘PVP-iodine has been reported to significantly reduce PPD.’

Response:

Many thanks for the comment. The suggestion has been implemented.

Revised text section:

“in 2010” was removed in Abstract Line 14

Comment 7: In the introduction, I will like the authors to stronger justification the rationale for conducting a review on PVP-iodine. The authors mentioned that CHX is the gold standard for oral antiseptics, what does PVP-iodine bring that is unique and therefore could be used for clinical practice, instead of using CHX?

Response:

We have gladly strengthened and emphasized the justification for PVP-iodine in the revised introduction and discussion section of our manuscript.

Revised text section:

Page 2 Line 62: The main advantage of PVP-Iodine, in contrast to other antiseptics, is therefore attributed in the literature in particular to its broad spectrum of activity (Caufield, Allen, & Childers, 1987; Gocke, Ponticas, & Pollack, 1985; Schreier et al., 1997), while showing a low toxicity and few contraindications (Kunisada et al., 1997; Maruniak et al.; 1992). The small molecule iodine can easily penetrate microorganisms and lead to their cell death through the oxidation of various substances (Lachapelle et al., 2013; McDonnell & Russell, 1999). PVP-Iodine showed a strong bacterial effect on clinical isolates even at a low concentration of 0.1% (Gocke et al., 1985). Its range of action covers gram-positive and gram-negative bacteria (Schreier et al., 1997), a wide range of enveloped and nonenveloped viruses (Kawana et al., 1997; Wutzler, Sauerbrei, Klocking, Brogmann, & Reimer, 2002), spores (Lachapelle et al., 2013) and fungal biofilms (Capriotti et al., 2018).

Page 2 Line 73: “After a systematic review by our group more than 10 years ago had shown an additional benefit of the use of PVP iodine (Sahrmann et al., 2010), and taking into account the new randomized clinical trials published in the past few years, we felt that a careful update of the evidence for the use of PVP iodine in step II periodontal therapy was necessary. Therefore, the aim of the present study was to re-evaluate the available literature and to assess the potential effect of the adjunctive use of PVP iodine on the clinical outcomes of periodontal therapy.”

Comment 8: Line 88-89 – it may be more helpful to use PMPR or periodontal instrumentation, instead of scaling root planing, in line with S3 guidelines.

Response:

Adapting the term used for the new classification we changed the term to “periodontal instrumentation” throughout the manuscript.

Revised text section:

(“Scaling and root planing” was replaced by “periodontal instrumentation” throughout the manuscript)

Comment 9: Line 111 – regarding excluded studies. I read through Supplemental Table II and noticed a lot of the studies were excluded because of ‘multiple publication of data from identical cohort’. What does this mean? If the authors are referencing another study that was included in the review, please state which studies were this in the supplement table in the ‘exclusion criteria’ column as well. Also, the full reference list of the excluded studies should be appended at the end of the document.

Response:

In the revised manuscript, the authors have replaced the term "multiple publication of data from identical cohort" by the more comprehensible term “conference paper”. In addition, we have referred to respective article that was included to the review.

Revised text section:

(Sources from excluded studies were added at the end of Supplements + adapted Supplement table II with more comprehensible term)

Comment 10: Line 115 – reference missing

Response:

We would like to apologize for the inaccuracy. A hyperlink was not recognized correctly when the work was transferred into the pdf. We have now corrected this.

Revised text section:

(Correct reference has been inserted and the hyperlink has been adjusted)

Comment 11: Table 1. Gender percentage and mean age – should include SDs, if reported. Also, what strikes me is that some of the studies have disproportionate numbers in terms of gender. For example, Al-Saeed 09 has 38% female, and Sahrmann 14 has 18.2% females – why the disproportionate genders, especially if these were RCTs?

Response:

The authors appreciate the detailed feedback on the included RCTs. We have now added the SDs wherever they were reported. Unfortunately, this was only the case in 2 studies. With regard to the gender distribution, it is important that both studies mentioned by the reviewer show a split-mouth design. No evidence was found that antiseptics had a strong gender-specific effect. Accordingly, the results seem still valid to the authors.

Revised text section:

(Adjusted SDs in Table 1)

Comment 12: Page 6 Line 6 – reference missing

Response:

We would like to apologize for the inaccuracy. A hyperlink was not recognized when the work was transferred into the pdf. The hyperlink was now adapted.

Revised text section:

(Correct reference has been inserted and the hyperlink has been adjusted)

Comment 13: Page 6 Line 5 – for GRADE, there are four classes (high, moderate, low, very low). Why did the authors exclude ‘very low’ from the classification?

Response:

Again, the authors want to thank for your valuable comment. As none of the studies was rated as “very low”, we missed this category in the description. We would like to apologize for this. In the revised version, we have added the very low rating, even though it was not assigned to any of the studies.

Revised text section:

In chapter 2.6: “in four different degrees (high, medium, low and very low) “, “very low (<2)” and in description of table II: “<2= very low quality”

Comment 14: Table 2 – Concealement is a typographical error.

Response:

Many thanks for the correction. This has been corrected in the table.

Revised text section:

“Concealement” was changed to “Concealment” in table 2.

Comment 15: Regarding statistical analysis – some of the included studies were split-mouth studies – did the meta-analysis account for clustered data by comparing the variance within clusters (ie the same patient?)

Response:

The reviewers would like to thank you for your statistical contribution. Unfortunately, the published data of the respective studies did not allow for patient-specific sub-analyses, so the meta-analysis does not account for clustered data.

Revised text section:

n/a

Comment 16: Page 8 Line 18 – reference missing

Response:

We would like to apologize for the inaccuracy. Again, the hyperlink was not recognized correctly when the work was transferred. We have now corrected this.

Revised text section:

(Correct reference has been inserted and the hyperlink has been adjusted)

Comment 17: Page 8 Line 29 – ‘one study lacked proper data reporting’. Were the authors of that study contacted for further information?

Response:

Thank you very much for the critical question. We had written to the respective correspondence address, but so far, we have not received any feedback. We have now clarified the point in the text.

Revised text section:

In chapter 3.1: “while in one study mean values/standard deviations of the relevant parameters were not reported. Unfortunately, it was not possible to contact the authors to obtain this data.”

Comment 18: Page 9 Line 35 – reference missing

Response:

Same problem: Again, we would like to apologize for the inaccuracy. A hyperlink was not recognized correctly when the work was transferred. We have now corrected this.

Revised text section:

(Correct reference has been inserted and the hyperlink has been adjusted)

Comment 19: Page 9 Line 45 –the ‘classification of 2018’ is not very precise. It may be better to label it as the ‘2018 AAP/EFP classification of periodontal diseases’.

Response:

Thank you for the specification. We have gladly adopted them.

Revised text section:

“classification of 2018” was changed to “2018 AAP/EFP classification of periodontal diseases”

Comment 20: Page 10 Line 81, 92 – reference missing

Response:

We would like to apologize for the inaccuracy. A hyperlink was not recognized correctly when the work was transferred. We have now corrected this.

Revised text section: (Correct reference has been inserted and the hyperlink has been adjusted)

Comment 21: I appreciate the funnel plot to assess publication bias, but with such a low number of studies included in this review, it is of limited use.

Response:

The authors appreciate the reviewer’s input and share the point of view. Though the benefit might be limited the respective blot nevertheless offers additional information to the reader, especially with regard to the study by do Vale et al. (please refer to the revised discussion section). In the revised manuscript we referred to the limited relevance of the funnel plot.

Revised text section:

Page 14, Line 170: “although the relevance is limited due to the small number of included studies”

Reviewer 2 Report

Comments and Suggestions for Authors

Dear Authors,

The manuscript entitled "Effects of Rinsing with Povidone-Iodine during step II Periodontal Therapy: A Systematic Review and Meta-Analysis" is a metaanalysis which  systematically re-assess the previously literature regarding a potential benefit of PVP-iodine in step II periodontal therapy.

The article is clear, well-written, relevant to the field, and has no weak points. I recommend its acceptance after minor revision.

Here are my suggestions:

- The first column of Table 1 should contain the authors and year of publication needs revision in the whole table. And in the 5th column, 6th and 7th line should be replaced with n/a or missing data.

- The statement "Error! Reference source not found." repeats in lines 115 page 3, 7 page 6, 18 page 8,35 page 9,line 92 page 10,

- In the Prisma chartflow(page 9) is there any use to mention in the Identification zone: Records removed for other reason n=0???

Best regards!

Author Response

Initial comment:

The manuscript entitled "Effects of Rinsing with Povidone-Iodine during step II Periodontal Therapy: A Systematic Review and Meta-Analysis" is a metaanalysis which systematically re-assess the previously literature regarding a potential benefit of PVP-iodine in step II periodontal therapy.

The article is clear, well-written, relevant to the field, and has no weak points. I recommend its acceptance after minor revision.       

Response:

The authors would like to thank the reviewer for the kind appreciation and the provided feedback.

Revised text section:

n/a

Comment 1: The first column of Table 1 should contain the authors and year of publication needs revision in the whole table. And in the 5th column, 6th and 7th line should be replaced with n/a or missing data.

Response:

The authors thank for the remark. We have revised and supplemented the table according to the reviewer’s suggestions

Revised text section:

(The year of publication and the authors has been supplemented in full to Table 1. In addition, n/a was noted in rows 6 and 7 in column 5.)

Comment 2: The statement "Error! Reference source not found." repeats in lines 115 page 3, 7 page 6, 18 page 8, 35 page 9, line 92 page 10.

Response:

We would like to apologize for the inaccuracy. A hyperlink was not recognized correctly when the work was transferred. We have now corrected this.

Revised text section: (Correct reference has been inserted and the hyperlink was adjusted)Comment 3.  In the Prisma chartflow (page 9) is there any use to mention in the Identification zone: Records removed for other reason n=0???

Response:

Thank you very much for the comment. We have removed this unnecessary line.

Revised text section:

(“records removed for other reason” was removed from the Prisma flowchart)

Reviewer 3 Report

Comments and Suggestions for Authors

Dear authors,

I suggest to improve the introduction and discussion with PVP - Iodine mode of action.

There are some errors about references :

- pag 6 line 7

- pag 8 line 18

- pag 9 line 35

- pag 10 line 81 and 92

Author Response

Dear authors,

I suggest to improve the introduction and discussion with PVP - Iodine mode of action.

Response:

The authors would like to thank you very much for your comment. We have now added a section dealing with the mode of action of PVP-iodine. In addition, we have expanded the introduction and discussion in several sections.

Revised text section:

Page 2 Line 62: The main advantage of PVP-Iodine, in contrast to other antiseptics, is therefore attributed in the literature in particular to its broad spectrum of activity (Caufield, Allen, & Childers, 1987; Gocke, Ponticas, & Pollack, 1985; Schreier et al., 1997), while showing a low toxicity and few contraindications (Kunisada et al., 1997; Maruniak et al.; 1992). The small molecule iodine can easily penetrate microorganisms and lead to their cell death through the oxidation of various substances (Lachapelle et al., 2013; McDonnell & Russell, 1999). PVP-Iodine showed a strong bacterial effect on clinical isolates even at a low concentration of 0.1% (Gocke et al., 1985). Its range of action covers gram-positive and gram-negative bacteria (Schreier et al., 1997), a wide range of enveloped and nonenveloped viruses (Kawana et al., 1997; Wutzler, Sauerbrei, Klocking, Brogmann, & Reimer, 2002), spores (Lachapelle et al., 2013) and fungal biofilms (Capriotti et al., 2018).

Page 14 Line 200 (Discussion): Nevertheless, it must be mentioned, particularly with regard to clearance, that even high initial PVP-iodine concentrations, are washed out and diluted after a short time (with saliva). It has been shown that the concentration of a 10% PVP-iodine rinse was reduced by just around 50% after only 1 minute and after 15 minutes only around 10% of the initial PVP-iodine concentration remained (Sahrmann, Manz, Attin, Zbinden, & Schmidlin, 2015). It can therefore be surmised that the concentration is certainly relevant, however due to the rapid dilution time of PVP-Iodine (Sahrmann, Manz, Attin, Zbinden, & Schmidlin, 2015), the far greater part depends on the type and above all the duration of application. In addition, it must be mentioned that a concentration of 0.1% PVP-Iodine is already able to kill a large number of clinical isolates (Gocke et al., 1985).

Comment 1: There are some errors about references:

- pag 6 line 7

- pag 8 line 18

- pag 9 line 35

- pag 10 line 81 and 92

Response:

We would like to apologize for the inaccuracy. A hyperlink was not recognized correctly when the work was transferred. We have now corrected this.

Revised text section:

(Correct reference has been inserted and the hyperlink was adjusted)

Round 2

Reviewer 1 Report

Comments and Suggestions for Authors

The authors have adquately addressed the issues raised and is fit for publication.

Comments on the Quality of English Language

Minor adjustments required.